# Anemia biomarkers and mortality in hemodialysis patients with or without diabetes: A 10-year follow-up study

Jihane Asmar[1], Dania Chelala[2,3], Razane El Hajj Chehade[1], Hiba Azar[2,3], Serge Finianos[2,3], Mabel Aoun[2,4]*

1 Department of Internal Medicine, Faculty of Medicine, Saint-Joseph University, Beirut, Lebanon, 2 Department of Nephrology, Faculty of Medicine, Saint-Joseph University, Beirut, Lebanon, 3 Department of Nephrology, Hotel-Dieu de France Hospital, Beirut, Lebanon, 4 Department of Nephrology, Saint-George Hospital, Ajaltoun, Lebanon

* aounmabel@yahoo.fr

**Data Availability Statement:** All relevant data are within the paper and its Supporting information files.

## Abstract

### Background

Many studies have assessed the association between anemia and mortality in hemodialysis but few compared patients with and without diabetes. Our study aims to investigate the impact of hemoglobin and iron parameters on mortality in hemodialysis patients with or without diabetes.

### Methods

This is a two-center retrospective study that included all adult patients who started hemodialysis between February 2012 and February 2020, followed until January 2021. Averages of hemoglobin, ferritin and transferrin saturation of entire follow-up were recorded. Kaplan Meier survival, log rank test and cox regression analyses were performed to assess the association between anemia biomarkers and mortality.

### Results

A total of 214 patients were included. Mean age was 67.98 ±12.41 years, mean hemoglobin was 10.92 ±0.75 g/dL, mean ferritin was 504.43 ± 221.42 ng/mL and mean transferrin saturation was 26.23 ±7.77%. Log rank test showed an association between hemoglobin ≥11 g/dL and better survival in patients without diabetes (P = 0.028). Based on cox regression analysis, hemoglobin was associated with all-cause mortality in all patients (HR = 0.66; CI:0.49,0.89; P = 0.007). When comparing patients with and without diabetes, this association remained significant only in patients without diabetes (HR = 0.53; CI:0.37,0.77; P<0.001). Based on different multivariate models, hemoglobin, ferritin and age were independent factors associated with mortality in patients without diabetes.

**Funding:** The authors received no specific funding for this work.

**Competing interests:** The authors have declared that no competing interests exist.

**Abbreviations: CAD**, coronary artery disease; **CI**, confidence interval; **CKD**, chronic kidney disease; **ESKD**, end-stage kidney disease; **Hb**, hemoglobin; **HD**, hemodialysis; **HR**, hazard ratio; **TSAT**, transferrin saturation.

## Conclusions

This study showed that hemoglobin ≥11 g/dL is associated with better survival in hemodialysis patients without diabetes but not in those with diabetes. These differences need to be further explored in other countries and settings. An individualization of the hemoglobin target level might be necessary to improve patients' outcomes.

## Introduction

Since 1990, chronic kidney disease (CKD) has been recognized as an important contributor to the Global Burden of Disease [1]. The ascending burden of CKD differs between a country and another and it depends on the ageing of populations, the increase in diabetes, obesity, hypertension and the survival of patients with end-stage kidney disease (ESKD) requiring kidney replacement therapy (KRT) [1, 2]. It is globally estimated that 69% of patients with ESKD are treated with hemodialysis (HD) [3]. Therefore, it is very important to study the modifiable factors that affect mortality of HD patients in each country and context.

Anemia is one of the major factors that influence HD patients' survival as confirmed by many studies [4–10]. In 1999, a large Medicare study revealed that a hematocrit between 33 and 36% was linked to the best survival in HD patients [5]. In 2004, the Dialysis Outcomes and Practice Patterns Study (DOPPS) group studied HD patients from 5 European countries and suggested to keep hemoglobin (Hb) above 11 g/dL [7]. Later, the Treat to Reduce Cardiovascular Events with Aranesp Therapy (TREAT) trial revealed more harm in high levels of Hb and influenced the international guidelines on anemia to recommend a hemoglobin of 11 to 12 g/dL in all patients CKD dialysis and non-dialysis dependent, with or without diabetes [11–13]. Based also on the DOPPS studies from the US and Europe, Hb is considered as one of the six modifiable variables that may lead to improvement of survival with adequate interventions in HD patients [14, 15]. They lowered their target of Hb from above 11 to above 10 g/dL based on the latest guidelines and showed that this would contribute to a gain in life-years [15]. However, there was no distinction in their studies between patients with and without diabetes.

Anemia is not only assessed and managed in HD patients based on Hb but also on ferritin and transferrin saturation (TSAT) [12]. However, the association of ferritin and TSAT levels with mortality is more controversial. In 2019, the PIVOTAL study showed less morbidity and mortality when HD patients were administered high doses of intravenous (IV) iron and achieved a median ferritin level of 700 ng/mL [16]. Despite these findings, there are still concerns regarding the infection-related mortality with high iron intake and high levels of ferritin and TSAT [13].

Based on the published literature, we identified three gaps that need to be addressed. First, it is not clear so far whether HD patients without diabetes should be recommended higher Hb levels. Very recently, Maruyama et al in Japan followed HD patients for one year and found out a relation between Hb and mortality only in patients without diabetes, a finding that has been only reported ten years ago by the DOPPS group in Japan [17, 18]. Second, most of the studies on anemia and mortality analyzed one level of Hb or ferritin in time or- as a trend lately- the variability of several levels of Hb or iron parameters [19–24]. However, to our knowledge, no previous study has evaluated the impact of an average level of anemia biomarkers estimated from the entire follow-up. Third, data on anemia and mortality in HD patients are numerous from Europe, the US and Japan but are very scarce in countries from the Eastern Mediterranean region. Therefore, our present study aims to investigate the relation between

hemoglobin and iron parameters with mortality in Lebanese HD patients with and without diabetes.

## Materials and methods

### Study design, setting and participants

This is a retrospective study that included all consecutive patients initiating hemodialysis in two Lebanese dialysis centers between February 2012 and February 2020. The follow-up period ranged between February 2012 and January 2021. The two dialysis units are located in two distinct governorates: Saint-George Hospital (HSGA) in Ajaltoun, Mount Lebanon and Hotel-Dieu de France Hospital (HDF) in Beirut. The six nephrologists of these two units have been following the KDIGO guidelines for anemia management. The erythropoietin stimulating agent (ESA) used in the two units is the short-acting recombinant human erythropoietin (rHuEPO).

All dialysis patients > 18 years old who underwent chronic hemodialysis for at least 3 months were included. We excluded any patient requiring multiple transfusions (at least one packed red blood cell every 2 months), patients who did not have a ferritin and transferrin saturation (TSAT) levels in their records throughout their follow-up, those who were transferred to another hospital, lost to follow-up, received a kidney transplant, shifted to peritoneal dialysis, suffering from hemolysis, myelodysplasia or chronic inflammatory bowel disease.

### Data collection

The variables were collected from the medical records of the two dialysis units. They included age, sex, cause of end-stage kidney disease, dialysis vintage, vascular access (arteriovenous fistula, graft or catheter), frequency of dialysis weekly, diabetes, hypertension, current smoking status, living at an altitude above 1000 meters, coronary artery disease (CAD) defined as a history of coronary artery stenosis, coronary artery bypass graft, stent or myocardial infarction, chronic obstructive pulmonary disease (COPD), episodes of bleeding during follow-up, medications' intake such as antiplatelet therapy, renin-angiotensin system inhibitors (RAASi), the dose of rHuEPO and the dose of intravenous (IV) iron in number of vials (one vial equals 100 mg of iron sucrose). The laboratory parameters included Hb, ferritin, TSAT, platelets' count, parathyroid hormone (PTH), serum albumin and urea reduction ratio (URR).

### Measurements

The laboratory biomarkers including Hb, ferritin, TSAT (iron/TIBC), urea, PTH, platelets and serum albumin were measured using standard techniques in the two hospitals.

For all patients, Hb, platelets and URR were estimated by computing the average of all monthly measurements available on the system during the total follow-up. Ferritin and TSAT were estimated by computing the average of all ferritin and TSAT measurements, available at a frequency of two to three times per year in all records across all follow-up. The PTH was also estimated as an average of all PTH of the entire follow-up with PTH levels of HDF unit multiplied by two because they used the third-generation technique.

### Outcomes

The main outcome of this study was all-cause mortality. We extracted death, date of death and cause of death from the medical records of patients in the two units. Sudden cardiac death, heart failure, myocardial infarction, severe aortic stenosis, aortic dissection, ischemic stroke and vascular ischemia were considered as causes for cardiovascular death. Infection-related

mortality did not include COVID-19 cases. The other causes of death collected were: cancer, bleeding (intracranial, thoracic, digestive and retroperitoneal), respiratory arrest and trauma.

### Statistical analysis

Categorical variables are presented as numbers and percentages. Continuous variables are reported as mean ± standard deviation (SD) if normally distributed and as medians with inter-quartile range (IQR) for variables not normally distributed. The independent t test was used to compare two continuous variables with normal distribution and Mann Whitney U test was used to compare skewed continuous data. The chi-square test was used to compare categorical variables between two groups. Hemoglobin was assessed as a continuous variable in all analyses and as a categorical variable in the survival analysis by dividing patients into two groups based on the level of Hb $< 11$ or $\geq 11$ g/dL. The threshold of 11 g/dL was selected taking into account that the target of hemoglobin in the latest guidelines varied between 10 and 12 g/dL.

Kaplan Meier survival analysis was performed to draw the curves of survival of two groups, with and without diabetes. Log rank, Breslow and Tarone Ware tests were used to compare the survival between those with low and high hemoglobin levels. Hazard ratios (HRs) and 95% confidence intervals (CIs) for all-cause mortality were assessed using cox regression analysis. Cox proportional hazard regression models were used to assess the independent association of hemoglobin with all-cause mortality.

There was 13.1% missing data for the IV irone dose and 8.1% missing data for the EPO dose; the multiple imputation regression model was used to replace the missing values. The imputed data was used to analyze the multivariable cox regression model including EPO and IV iron doses. Statistical analyses were performed using the Statistical Package for Social Sciences (SPSS), version 25.0. A *P*-value $<0.05$ was considered statistically significant.

### Ethical considerations

The study got the approval of the ethics committee of Hotel-Dieu de France-affiliated to Saint-Joseph University (ID Tfem/2022/9) and was conducted in agreement with the Helsinki Declaration of 1975. Since it was a retrospective study, the participant consent was waived by the Saint-Joseph University ethics committee (Tfem/2022/9).

## Results

### General characteristics

The flow diagram summarizes the process of inclusion of eligible patients (S1 Fig). A total of 214 hemodialysis patients were finally analyzed. Table 1 depicts their general characteristics and the comparison between patients with and without diabetes. Their mean age was 67.98 ±12.41 years and 64% were men. The follow-up of patients ranged between a minimum and a maximum of 4 and 107 months respectively; only 5 patients had a follow-up of less than 6 months. Around 23% of patients underwent twice-weekly hemodialysis (all of them from one unit). Their mean hemoglobin was 10.92 ±0.75 g/dL, the median ferritin was 514.5 (360.0, 623.7) ng/mL and the mean TSAT was 26.23 ±7.77%.

### Anemia biomarkers and other laboratory results

The mean hemoglobin of all patients was 10.92±0.75 g/dL, the mean ferritin was 504.43 ±221.42 ng/mL and the mean TSAT was 26.23 ±7.77%. Table 2 shows the difference between patients with and without diabetes.

**Table 1. Comparison between patients with and without diabetes.**

| | Total sample N = 214 | Patients with diabetes n = 110 | Patients without diabetes n = 104 | *P* |
|---|---|---|---|---|
| **General characteristics and cardiovascular risk factors** | | | | |
| **Age at dialysis initiation, years Mean ±SD** | 67.98 ±12.41 | 68.52 ±10.79 | 67.4 ±13.95 | 0.516* |
| **Gender (M/F), n(%)** | 137/77 (64/36) | 70/40 (63.6/36.4) | 67/37 (64.4/35.6) | 0.905** |
| **Living at an altitude >1000 m, n(%)** | 45 (22.2) | 28 (27.2) | 17 (16.3) | 0.081** |
| **Smoking, n(%)** | 87 (40.7) | 47 (42.7) | 40 (38.5) | 0.525** |
| **Hypertension, n(%)** | 209 (97.7) | 109 (99.1) | 100 (96.2) | 0.202**** |
| **CAD, n(%)** | 103 (48.1) | 68 (61.8) | 35 (33.7) | <0.001** |
| **COPD, n(%)** | 62 (29) | 37 (33.6) | 25 (24.0) | 0.093** |
| **Dialysis characteristics** | | | | |
| **Cause of ESKD, n(%)** | | | | |
| Diabetes | 100 (46.7) | | | |
| Nephrosclerosis | 35 (16.4) | | | |
| Glomerular disease | 20 (9.3) | | | |
| Unknown origin | 17 (7.9) | | | |
| Polycystic kidney disease | 13 (6.1) | | | |
| One kidney | 9 (4.2) | | | |
| Tubulointerstitial nephritis | 8 (3.7) | | | |
| Post-Obstructive | 4 (1.9) | | | |
| Others | 8 (3.7) | | | |
| **Dialysis vintage, months Mean ±SD** | 39.87 ±24.47 | 40.67 ±22.78 | 39.03 ±26.22 | 0.624* |
| **Three/Two sessions weekly, n(%)** | 165/49 (77.1/22.9) | 92/18 (83.6/16.4) | 73/31 (70.2/29.8) | 0.019** |
| **Catheter as vascular access, n(%)** | 24 (11.2) | 13 (11.8) | 13 (10.6) | 0.776** |
| **History of bleeding, n(%)** | 57 (26.6) | 29 (26.4) | 28 (26.9) | 0.570** |
| **Medications** | | | | |
| **Aspirin or clopidogrel or anticoagulant intake, n(%)** | 131 (61.2) | 82 (74.5) | 49 (47.1) | <0.001** |
| **RAASi intake, n(%)** | 29 (13.6) | 12 (10.9) | 17 (16.3) | 0.234** |
| **rHuEPO intake, UI/month *(After imputation)* Mean ±SE** | 33097.8 ±1680.3 | 34303.7 ±2411.3 | 31822.3 ± 2332.3 | 0.460* |
| **IV Iron intake, vials/month *(After imputation)* Mean ±SE** | 1.44 ±0.05 | 1.52 ±0.07 | 1.36 ±0.07 | 0.098* |
| **Outcome** | | | | |
| **Death, n(%)** | 83 (38.8) | 50 (45.5) | 33 (31.7) | 0.039** |
| **Cause of death, n(%)** | | | | 0.916** |
| • Cardiovascular | 38 (48.1) | 25 (49) | 15 (42.9) | |
| • Infection | 23 (29.1) | 16 (31.4) | 10 (28.6) | |
| • Cancer | 5 (6.3) | 2 | 3 | |
| • Bleeding | 2 (2.5) | 1 | 2 | |
| • COVID-19 | 4 (5.1) | 3 | 2 | |
| • Other causes | 7 (8.9) | 4 | 3 | |

Note.

*t independent test

**Chi Square test

***Mann Whitney U test

****Fischer's Exact test

Abbreviations. CAD, coronary artery disease; COPD, chronic obstructive pulmonary disease; RAASi, Renin-angiotensin-aldosterone-system inhibitors

**Table 2. Laboratory tests: Comparison of patients with and without diabetes.**

| | Total N = 214 | With diabetes n = 110 | Without diabetes n = 104 | *P* |
|---|---|---|---|---|
| **Hemoglobin, g/dL Mean ±SD** | 10.92 ± 0.75 | 10.91 ±0.72 | 10.93 ±0.79 | 0.827* |
| Hb<10, n(%) | 23 (10.7) | 12 (11.5) | 11 (10) | |
| Hb 10–10.99, n(%) | 90 (42.1) | 42 (40.4) | 48 (43.6) | 0.730*** |
| Hb 11–11.99, n(%) | 84 (39.3) | 43 (41.3) | 41 (37.3) | |
| Hb ≥12, n(%) | 17 (7.9) | 7 (6.7) | 10 (9.1) | |
| **Ferritin, ng/mL, Mean ±SD** | 504.43 ± 221.42 | 477.75 ± 206.88 | 532.64 ± 233.52 | 0.071* |
| Ferritin< 500, n(%) | 105 (49.1) | 59 (53.6) | 46 (44.2) | |
| Ferritin 500–800, n(%) | 90 (42.1) | 44 (40) | 46 (44.2) | 0.246*** |
| Ferritin>800, n(%) | 19 (8.9) | 7 (6.4) | 12 (11.5) | |
| **TSAT, %, Mean ±SD** | 26.23 ±7.77 | 26.4 ±8.2 | 26.1 ±7.3 | 0.789* |
| TSAT<30, n(%) | 155 (72.4) | 80 (72.7) | 75 (72.1) | |
| TSAT 30–50, n(%) | 58 (27.1) | 29 (26.4) | 29 (27.9) | |
| TSAT>50, n(%) | 1 (0.5) | 1 (0.9) | 0 | |
| **URR, Mean ±SD** | 0.74 ±0.05 | 0.73 ±0.05 | 0.74 ±0.05 | 0.087* |
| **PTH, pg/mL Median (IQR)** | 198.6 (121.4, 351.5) | 162.9 (106.1, 314.8) | 233.5 (138.2, 395.7) | 0.016** |
| **Serum albumin, g/dL, Mean ±SD** | 3.72 ±0.38 | 3.67 ±0.37 | 3.77 ±0.38 | 0.044* |
| **Platelets' count, 10⁹/l Mean ±SD** | 215.4 ±67.4 | 219.4 ±71.5 | 211.2 ±62.9 | 0.378* |

Note.

*t independent test

**Mann Whitney U test

***Chi-Square test

Abbreviations. TSAT, transferrin saturation; PTH, parathyroid hormone; URR, urea reduction ratio.

## Survival analysis

The Kaplan-Meier curves illustrate the effect of low and high hemoglobin on survival of patients with or without diabetes (S2 Fig, Figs 1 and 2). The median survival time of all patients was 63 months (95%CI, 51.7–74.3).

The log rank test depicted a significant association between higher hemoglobin and better survival in patients without diabetes with $P = 0.028$ (Fig 1). The association was also significant using the Breslow test ($P = 0.038$) and the Tarone-Ware test ($P = 0.031$).

## Cox regression analysis

An analysis of the total sample of patients showed age, hemoglobin, ferritin ≥ 800, PTH and serum albumin as factors associated with all-cause mortality (Table 3).

After dividing the sample into patients with and without diabetes, we found age, hemoglobin and serum albumin significantly associated with all-cause mortality in patients without diabetes (Table 4). In patients with diabetes, PTH level, ferritin ≥ 800 and serum albumin were significantly associated with higher mortality (Table 4).

## Cox proportional hazard regression models

Based on different multivariable models including age, serum albumin, hemoglobin, ferritin and TSAT, we found that hemoglobin, ferritin, serum albumin and age were independent factors associated with all-cause mortality in patients without diabetes (Table 5). Serum albumin was the only factor associated with mortality in both groups, diabetes and no diabetes.

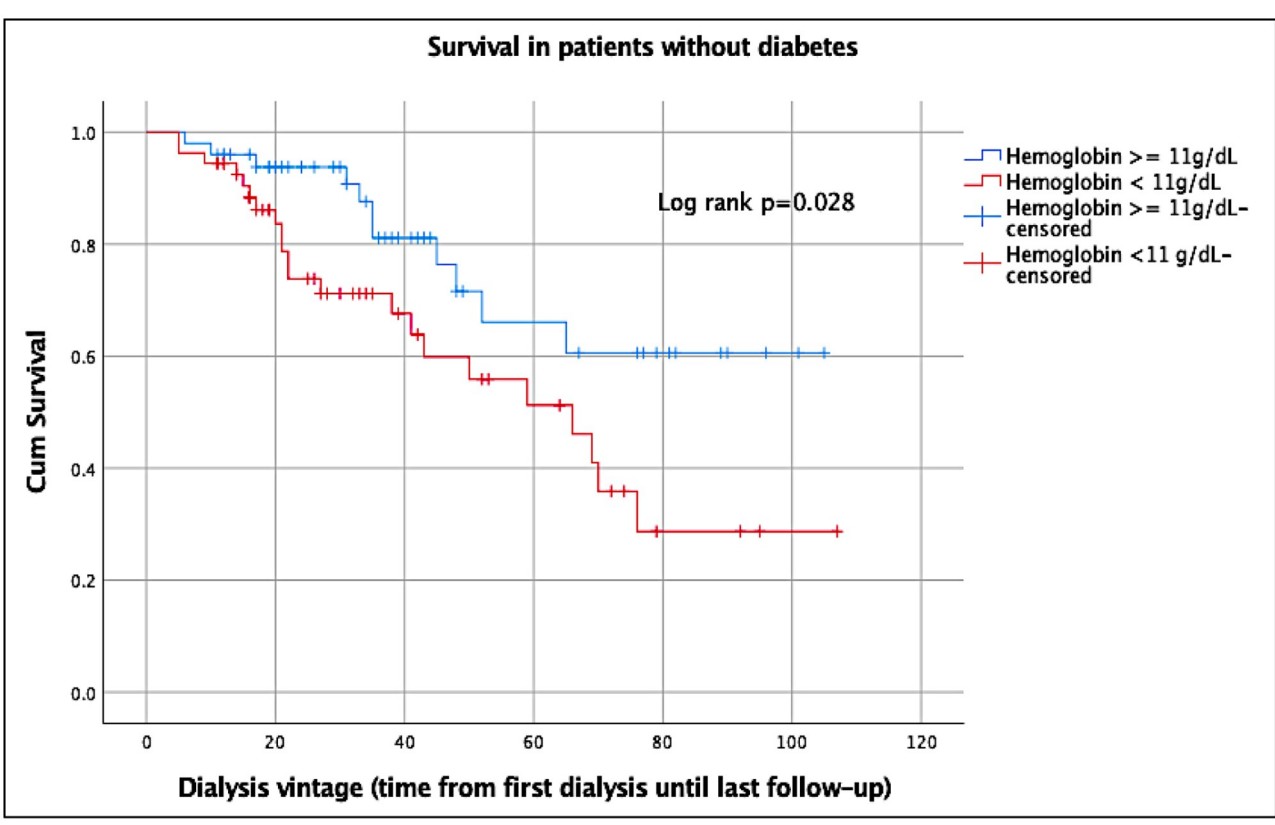

**Fig 1. Kaplan-Meier survival curve of patients without diabetes.** Note. Dialysis vintage in months.

### Factors associated with different causes of mortality

When comparing patients with infection-related mortality to other causes of death, we did not find an association between hemoglobin and mortality (HR = 0.75; 95%CI: 0.42,1.35; $P$ = 0.337). In patients with diabetes (but not in patients without diabetes) ferritin $\geq$ 800 ng/mL was associated with high risk of infection-related mortality (HR = 12.7; 95% CI:3.19, 50.52; $P$<0.001). Dialysis using a catheter versus fistula or graft increased the risk of infection-related mortality in patients without diabetes and with diabetes (HR = 4.72; 95% CI:1.18,18.94; $P$ = 0.028 and HR = 3.55; 95% CI:1.17, 10.79; $P$ = 0.025 respectively).

When comparing cardiovascular to non-cardiovascular mortality, age was found to be an independent predictor of non-cardiovascular mortality (Table 6).

### Discussion

Interestingly, this study did not show a significant association between hemoglobin and mortality in HD patients with diabetes after a mean follow-up of 40 months. However, in patients without diabetes, every decrease in 1 g/dL of Hb increased the risk of mortality by an average of 50% depending on confounders and Hb $\geq$11g/dL was linked with a better survival. The absence of association in patients with diabetes can be explained by their several comorbidities and by the fact that anemia becomes trivial among the more important factors like atherosclerosis, coronary disease and infections. Similar to our study, a recent large national study of 149,308 HD patients from Japan found no association between Hb and mortality in diabetic patients, a finding concurrent with the J-DOPPS in 2012 [17, 18]. However, their cut-off of Hb

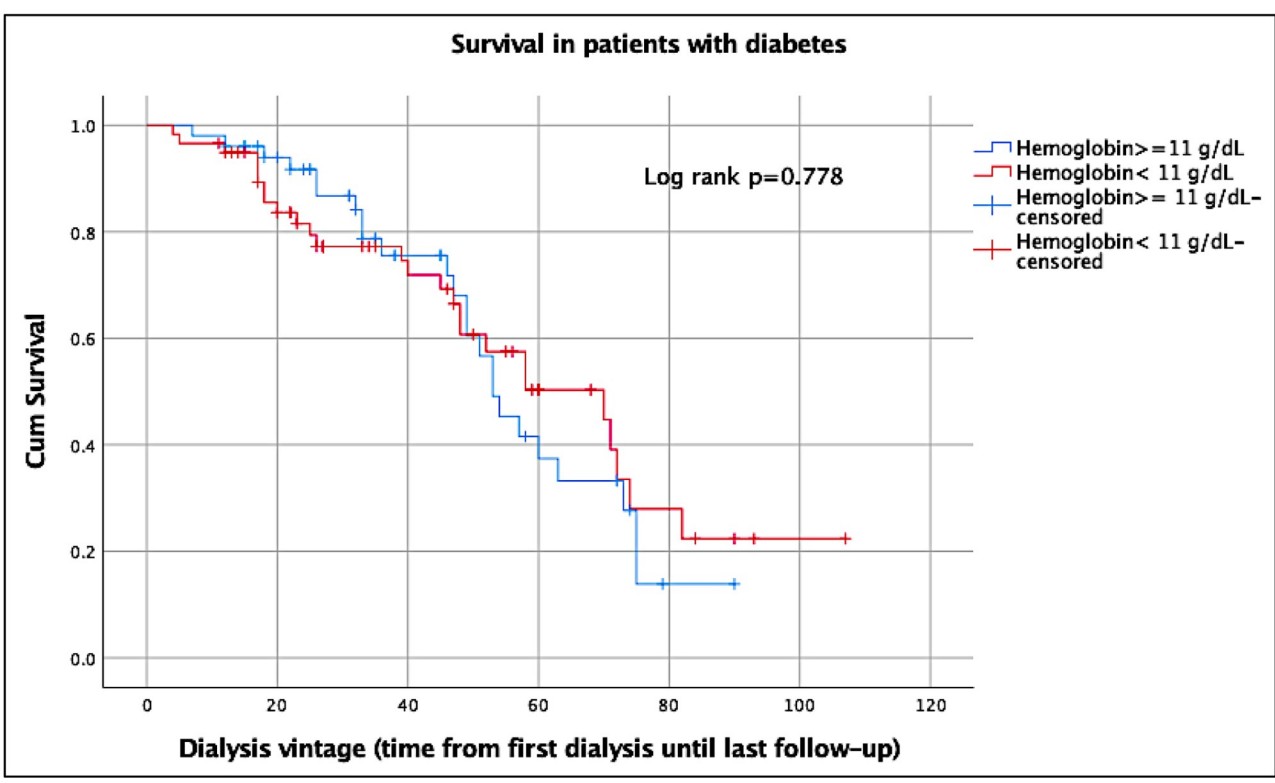

**Fig 2. Kaplan-Meier survival curve of patients with diabetes.** Note. Dialysis vintage in months.

was 9 g/dL much lower than the 11 g/dL of our study. Their methodology was also different. They collected one level of Hb and then followed the patients for one year. Our study has a longer follow-up and Hb reflected the average of the whole follow-up of patients. In alignment with our results, Villain et al from France [25] and Hanafusa et al from Japan [26] did not find an association between Hb and mortality in a group of elderly patients (in their 80s) and they suggested that elderly may better tolerate low Hb levels. But this cannot explain the lack of association in our sample of patients with diabetes who had a mean age of 68 years old.

Another important finding in our study is the association of ferritin levels with all-cause mortality only in the group of patients without diabetes and after adjustment to age and serum albumin. As for diabetes, we found a higher risk of infection-related mortality when ferritin was ≥ 800 ng/mL but the number of patients in this category was small making it hard to draw strong conclusions. In addition, we could not identify any association between TSAT levels and mortality. Many previous studies tried to clarify the relation between iron parameters, iron doses and death [16, 23, 27–31]. Some suggested that exceeding 400 mg of IV iron per month increases mortality and others showed better survival with high doses of iron [30, 31]. A large study from Taiwan showed a better survival when patients have a Hb >10 g/dL, a ferritin between 300 and 800 ng/mL and a TSAT between 30 and 50% [28]. Kuragano and al. showed that a high ferritin level increases the morbidity and mortality rate [23]. But all these studies did not compare patients based on their diabetes status. McDougall et al conducted the PIVOTAL trial that included 45% of diabetic patients and showed better survival with high doses of intravenous iron up to a median ferritin of 700 ng/mL [16]. And very recently, Karaboyas and al. analyzed the association between ferritin levels and mortality in HD patients

**Table 3. Cox regression analysis of factors associated with all-cause mortality in the total sample.**

| Variable | HR | 95%CI | P |
|---|---|---|---|
| Age, years | 1.04 | 1.02, 1.06 | 0.001 |
| Diabetes | 1.36 | 0.88, 2.12 | 0.169 |
| Living at an altitude >1000 m | 1.30 | 0.79, 2.15 | 0.307 |
| Smoking | 0.68 | 0.43, 1.05 | 0.083 |
| CAD | 1.38 | 0.88, 2.16 | 0.158 |
| COPD | 0.88 | 0.55, 1.42 | 0.612 |
| History of bleeding | 0.68 | 0.39, 1.17 | 0.160 |
| Aspirin intake | 1.47 | 0.91, 2.38 | 0.117 |
| RAASi intake | 0.51 | 0.25, 1.08 | 0.077 |
| Catheter as vascular access | 1.39 | 0.75, 2.57 | 0.298 |
| rHuEPO intake, UI/month | 1.00 | 1.00, 1.00 | 0.436 |
| IV Iron intake, vials/month | 1.49 | 0.94, 2.38 | 0.092 |
| Hemoglobin, g/dL | 0.66 | 0.49, 0.89 | 0.007 |
| Ferritin, ng/mL | 1.001 | 1.00, 1.002 | 0.199 |
| Ferritin≥800 (Ref:<800) | 3.02 | 1.53, 5.93 | 0.001 |
| TSAT, % | 1.012 | 0.98, 1.04 | 0.420 |
| PTH, pg/mL | 0.99 | 0.99, 0.99 | 0.005 |
| Serum albumin, g/L | 0.17 | 0.09, 0.29 | <0.001 |
| Platelets' count | 1.00 | 1.00, 1.00 | 0.667 |

Note. HR, Hazard Ratio; 95%CI, 95% Confidence Interval; CAD, coronary artery disease; COPD, chronic obstructive pulmonary disease; TSAT, transferrin saturation; PTH, parathyroid hormone.

**Table 4. Cox regression analysis of factors associated with all-cause mortality in two groups of patients with and without diabetes.**

| Variable | With diabetes | | | Without diabetes | | |
|---|---|---|---|---|---|---|
| | HR | 95%CI | P | HR | 95%CI | P |
| Age | 1.02 | 0.99, 1.05 | 0.252 | 1.08 | 1.03, 1.12 | <0.001 |
| Living at an altitude >1000 m | 1.35 | 0.72, 2.52 | 0.345 | 1.12 | 0.46, 2.72 | 0.804 |
| Smoking, current | 0.77 | 0.44, 1.34 | 0.350 | 0.60 | 0.29, 1.23 | 0.164 |
| CAD | 1.09 | 0.59, 2.00 | 0.789 | 1.59 | 0.80, 3.17 | 0.184 |
| COPD | 0.91 | 0.49, 1.67 | 0.768 | 0.88 | 0.41, 1.91 | 0.747 |
| History of bleeding | 0.55 | 0.25, 1.17 | 0.119 | 0.94 | 0.43, 2.07 | 0.878 |
| Aspirin intake | 1.51 | 0.77, 2.97 | 0.229 | 1.47 | 0.71, 3.01 | 0.298 |
| RAASi intake | 0.57 | 0.22, 1.44 | 0.235 | 0.47 | 0.14, 1.55 | 0.218 |
| Catheter as vascular access | 0.94 | 0.39, 2.21 | 0.879 | 2.22 | 0.91, 5.39 | 0.078 |
| rHuEPO intake, UI/month | 1.00 | 1.00, 1.00 | 0.182 | 1.00 | 1.00, 1.00 | 0.861 |
| IV Iron intake, vials/month | 1.31 | 0.68, 2.49 | 0.411 | 1.85 | 0.95, 3.59 | 0.069 |
| Hemoglobin, g/dL | 0.93 | 0.59, 1.43 | 0.731 | 0.53 | 0.37, 0.77 | <0.001 |
| Ferritin, ng/mL | 1.001 | 0.99, 1.002 | 0.244 | 1.001 | 0.99, 1.002 | 0.447 |
| Ferritin≥800 (Ref:<800) | 3.95 | 1.53, 10.19 | 0.004 | 2.38 | 0.90, 6.26 | 0.079 |
| TSAT, % | 1.009 | 0.98, 1.04 | 0.569 | 1.011 | 0.96, 1.07 | 0.702 |
| PTH, pg/mL | 0.99 | 0.99, 1.00 | 0.023 | 0.99 | 0.99, 1.00 | 0.135 |
| Serum albumin, g/L | 0.16 | 0.07, 0.38 | <0.001 | 0.17 | 0.08, 0.35 | <0.001 |

Note. HR, Hazard Ratio; 95%CI, 95% Confidence Interval; CAD, coronary artery disease; COPD, chronic obstructive pulmonary disease; TSAT, transferrin saturation; PTH, parathyroid hormone.

**Table 5. Cox proportional hazard regression models for all-cause mortality in patients with diabetes and those without diabetes.**

| | With diabetes | | | Without diabetes | | |
|---|---|---|---|---|---|---|
| **Model 1** | **HR** | **95%CI** | ***P*** | **HR** | **95%CI** | ***P*** |
| Hemoglobin, g/dL | 0.99 | 0.62, 1.59 | 0.979 | 0.49 | 0.32, 0.76 | 0.001 |
| Ferritin, ng/mL | 1.001 | 0.99, 1.002 | 0.305 | 0.99 | 0.99, 1.001 | 0.454 |
| TSAT | 1.006 | 0.97, 1.04 | 0.727 | 1.02 | 0.96, 1.08 | 0.596 |
| **Model 2** | **HR** | **95%CI** | ***P*** | **HR** | **95%CI** | ***P*** |
| Hemoglobin, g/dL | 1.07 | 0.63, 1.81 | 0.802 | 0.46 | 0.28, 0.75 | 0.002 |
| Ferritin, ng/mL | 1.001 | 0.99, 1.003 | 0.282 | 0.99 | 0.99, 1.00 | 0.057 |
| TSAT | 1.01 | 0.98, 1.05 | 0.465 | 1.02 | 0.96, 1.09 | 0.514 |
| Age, years | 1.03 | 0.99, 1.06 | 0.127 | 1.08 | 1.03, 1.12 | 0.001 |
| Altitude above1000m | 1.33 | 0.68, 2.61 | 0.406 | 0.84 | 0.33, 2.16 | 0.721 |
| Smoking | 0.68 | 0.36, 1.29 | 0.236 | 0.66 | 0.30, 1.44 | 0.298 |
| **Model 3** | **HR** | **95%CI** | ***P*** | **HR** | **95%CI** | ***P*** |
| Hemoglobin, g/dL | 0.89 | 0.69, 1.13 | 0.342 | 0.44 | 0.36, 0.55 | <0.001 |
| Ferritin, ng/mL | 1.001 | 1.00, 1.001 | 0.041 | 0.99 | 0.99, 1.001 | 0.070 |
| TSAT | 1.002 | 0.99, 1.02 | 0.759 | 1.01 | 0.99, 1.04 | 0.378 |
| rHuEPO intake, UI/month | 1.00 | 1.00, 1.00 | 0.031 | 1.00 | 1.00, 1.00 | 0.060 |
| IV Iron intake, vials/month | 1.06 | 0.85, 1.32 | 0.616 | 1.47 | 1.13, 1.91 | 0.004 |
| **Model 4** | **HR** | **95%CI** | ***P*** | **HR** | **95%CI** | ***P*** |
| Hemoglobin, g/dL | 1.24 | 0.76, 2.04 | 0.393 | 0.58 | 0.34, 0.98 | 0.042 |
| Ferritin, ng/mL | 1.002 | 1.00, 1.003 | 0.075 | 0.99 | 0.99, 1.00 | 0.033 |
| TSAT, % | 1.00 | 0.97, 1.03 | 0.986 | 1.04 | 0.98, 1.11 | 0.212 |
| Age, years | 0.99 | 0.96, 1.02 | 0.492 | 1.06 | 1.03, 1.10 | 0.001 |
| Serum albumin, g/L | 0.12 | 0.05, 0.32 | <0.001 | 0.20 | 0.06, 0.66 | 0.008 |
| **Model 5** | **HR** | **95%CI** | ***P*** | **HR** | **95%CI** | ***P*** |
| Hemoglobin, g/dL | 1.15 | 0.70, 1.89 | 0.571 | 0.52 | 0.28, 0.94 | 0.030 |
| Ferritin, ng/mL | 1.001 | 1.00, 1.003 | 0.114 | 0.99 | 0.99, 1.00 | 0.030 |
| TSAT | 0.99 | 0.97, 1.03 | 0.883 | 1.04 | 0.98, 1.11 | 0.200 |
| Age, years | 0.98 | 0.95, 1.02 | 0.256 | 1.05 | 1.01, 1.09 | 0.007 |
| Serum albumin, g/L | 0.14 | 0.05, 0.38 | <0.001 | 0.20 | 0.06, 0.68 | 0.010 |
| CAD | 1.43 | 0.75, 2.75 | 0,280 | 1.63 | 0.73, 3.65 | 0.232 |
| PTH, pg/mL | 0.99 | 0.99, 1.00 | 0.125 | 1.00 | 0.99, 1.001 | 0.643 |

Note. HR, Hazard Ratio; 95%CI, 95% Confidence Interval; CAD, coronary artery disease; TSAT, transferrin saturation; PTH, parathyroid hormone.

Model 1 includes Hemoglobin, ferritin and TSAT; Model 2 includes Model 1 variables and age, altitude and smoking status; Model 3 includes Model 1 variables and EPO and Iron dose; Model 4 includes Model 1 with albumin and age; Model 5 includes Model 4 with PTH and CAD.

**Table 6. Cox proportional hazard regression model of factors associated with cardiovascular or non-cardiovascular mortality.**

| Variable | Cardiovascular mortality | | | Non-cardiovascular mortality | | |
|---|---|---|---|---|---|---|
| | **HR** | **95%CI** | ***P*** | **HR** | **95%CI** | ***P*** |
| **Age, years** | 1.01 | 0.98, 1.05 | 0.429 | 1.05 | 1.02, 1.09 | 0.003 |
| **Diabetes** | 1.53 | 0.68, 3.45 | 0.303 | 1.29 | 0.71, 2.33 | 0.662 |
| **Hemoglobin, g/dL** | 0.65 | 0.36, 1.16 | 0.144 | 0.65 | 0.42, 1.01 | 0.054 |
| Ferritin, ng/mL | 1.00 | 0.99, 1.002 | 0.887 | 1.00 | 0.99, 1.002 | 0.758 |
| TSAT, % | 1.01 | 0.96, 1.07 | 0.659 | 1.003 | 0.96, 1.05 | 0.895 |

Note. HR, Hazard Ratio; 95%CI, 95% Confidence Interval.

from the DOPPS data in three major regions of the world [27]. The median ferritin levels in Japan, Europe and the United States were found to be 83, 405 and 718 ng/mL respectively [27]. Elevated serum ferritin (relative to the median of each region) was shown to be a risk factor for mortality [27]. However, adjusting ferritin levels to markers of inflammation and malnutrition made the association not significant in Japan and only significant above 1000 ng/mL in US and Europe [27]. Like hemoglobin, the difference in results between their study and ours may be due to the different methodology for they took one baseline level of ferritin and TSAT and followed patients for one year. It is noteworthy that our levels of ferritin are close to those of Europe and within the range of the international guidelines. Our doses of IV iron, available for 86% of patients, are not very high and average 150 mg per month.

Other demographic and modifiable factors have been identified as associated with mortality. The DOPPS initiative in the US and 10 European countries identified six modifiable variables in hemodialysis: hemoglobin, serum albumin, dialysis dose, catheter use for dialysis, phosphate and interdialytic weight gain [14, 15]. It is well known since the 70s that age and poor nutritional status are associated with higher death in dialysis patients [32]. In our cohort, age was associated with mortality in patients without diabetes, similar to hemoglobin. On the other hand, our results emphasize the strong predictive value of serum albumin for all-cause mortality among all categories of hemodialysis patients. This is in agreement with many previous reports [33–36]. Owen et al showed that the risk of mortality is tripled when serum albumin is <3.5 g/dL [33]. Capelli et al showed that the combination of low hemoglobin and serum albumin negatively affects patients' prognosis [36]. However, it is controversial whether low serum albumin is due to poor nutrition. de Mutsert et al demonstrated that inflammation rather than malnutrition explains the association between albumin and mortality [35].

Our study has also shown an association between low PTH and mortality in patients with diabetes. Several studies have demonstrated that abnormal serum parathyroid hormone (PTH) levels are associated with increased morbidity and mortality [37, 38]. Block et al found higher mortality rate with PTH<150 pg/mL in unadjusted analysis and with PTH >600 pg/mL after adjustment for age and diabetes [37]. Another factor that was analyzed in our study is altitude. The adjustment of hemoglobin to altitude did not affect our results whether in the group with diabetes or without diabetes. This is in contrast to one report from Sibbel et al who found lower mortality and higher hemoglobin in patients living at high altitude levels [39].

Finally, and in alignment with results seen in other surveys, our study showed that diabetes is not always associated with a greater risk of mortality in hemodialysis patients. This could be explained by the large number of patients excluded from our study because of their short follow-up (deceased early), the majority of whom were diabetic. The absence of relation between diabetes and mortality in HD was reported by Myers et al [40]. Only patients with low systolic blood pressure showed an association between diabetes and mortality; similar to our study's design, they excluded from their cohort patients who were dialyzed for less than 150 days [40].

## Limitations and strengths

This study has some limitations. It is a retrospective study with some missing data on doses of ESA and IV iron. Moreover, there is a lack of data on inflammatory biomarkers like C-Reactive Protein, thus it was impossible to adjust ferritin to the latter in order to eliminate the bias of inflammation. On the other hand, our study has several strengths. First, and to our knowledge, it is the first in the Middle East region to explore the relationship between mortality and anemia biomarkers in hemodialysis patients. Second, it has a long follow-up. Third, the estimated levels of hemoglobin, ferritin, TSAT, albumin and URR represent the average of all measurements at a mean follow-up of 40 months.

## Conclusion

In conclusion, this study showed that a hemoglobin $\geq$11 g/dL is associated with better survival in hemodialysis patients without diabetes but not in patients with diabetes. Targeting a hemoglobin level above 11 g/dL in patients without diabetes needs to be further studied. An individualization of the hemoglobin target level might be necessary to improve survival in hemodialysis patients.

## Supporting information

**S1 Fig. Flow diagram of patients' inclusion based on their eligibility.**
(TIFF)

**S2 Fig. Kaplan-Meier survival curve of the total sample.**
(TIFF)

**S1 File.**
(PDF)

**S2 File.**
(PDF)

**S1 Dataset.**
(XLSX)

## Acknowledgments

Authors would like to thank all dialysis staff who helped with the data collection.

## Author Contributions

**Conceptualization:** Jihane Asmar, Mabel Aoun.

**Data curation:** Jihane Asmar, Dania Chelala, Razane El Hajj Chehade, Hiba Azar, Serge Finianos, Mabel Aoun.

**Formal analysis:** Jihane Asmar, Mabel Aoun.

**Investigation:** Jihane Asmar, Razane El Hajj Chehade, Hiba Azar, Serge Finianos, Mabel Aoun.

**Methodology:** Jihane Asmar, Mabel Aoun.

**Project administration:** Mabel Aoun.

**Resources:** Dania Chelala, Hiba Azar, Serge Finianos, Mabel Aoun.

**Software:** Mabel Aoun.

**Supervision:** Mabel Aoun.

**Validation:** Dania Chelala, Mabel Aoun.

**Visualization:** Mabel Aoun.

**Writing – original draft:** Jihane Asmar, Mabel Aoun.

**Writing – review & editing:** Jihane Asmar, Dania Chelala, Mabel Aoun.

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
