## [Decision Letter · Decision Letter 0]

26 Sep 2022

PONE-D-22-23532Anemia biomarkers and mortality in hemodialysis patients with or without diabetes : a 10-year follow-up studyPLOS ONE

Dear Dr. Aoun, Thank you for submitting your manuscript to PLOS ONE. After careful consideration, we feel that it has merit but does not fully meet PLOS ONE’s publication criteria as it currently stands. Therefore, we invite you to submit a revised version of the manuscript that addresses the points raised during the review process.

We look forward to receiving your revised manuscript.

Kind regards,

Donovan Anthony McGrowder, PhD., MA., MSc

Academic Editor

PLOS ONE

Journal Requirements:

3.We note that you have indicated that data from this study are available upon request. PLOS only allows data to be available upon request if there are legal or ethical restrictions on sharing data publicly. For more information on unacceptable data access restrictions, please see http://journals.plos.org/plosone/s/data-availability#loc-unacceptable-data-access-restrictions. 

Additional Editor Comments:

Dear Dr. Aoun,

Your manuscript “Anemia biomarkers and mortality in hemodialysis patients with or without diabetes : a 10-year follow-up study” has been assessed by our reviewers. They have raised a number of points which we believe would improve the manuscript and may allow a revised version to be published in PLOS ONE. Their reports, together with any other comments, are below.

If you are able to fully address these points, we would encourage you to submit a revised manuscript to PLOS ONE.

Best regards,

Dr. Donovan McGrowder

Reviewers' comments:

Reviewer's Responses to Questions

**Comments to the Author**

1. Is the manuscript technically sound, and do the data support the conclusions?

Reviewer #1: Yes

Reviewer #2: Partly

2. Has the statistical analysis been performed appropriately and rigorously? 

Reviewer #1: Yes

Reviewer #2: Yes

3. Have the authors made all data underlying the findings in their manuscript fully available?

Reviewer #1: Yes

Reviewer #2: Yes

4. Is the manuscript presented in an intelligible fashion and written in standard English?

Reviewer #1: Yes

Reviewer #2: Yes

5. Review Comments to the Author

Reviewer #1: Jihane Asmar et al. investigated the impact of Hb, ferritin, and TSAT on mortality of HD patients with or without diabetes. Log rank test showed an association between Hb ≥ 11 g/dL and better survival in patients without diabetes but not in patients with diabetes. Hb, ferritin and age were independent factors associated with mortality in patients without diabetes. Although the enrolled patients were relatively small numbers, however, the study has strength such as long-term follow-up and usage of the average data of all parameters of Hb during entire observation period.

Major comments

1. In page 6 lines 1-2, did the authors investigate the compliance rate for the KDIGO guideline for anemia management in the current investigation.

2. In page 6 line 3, are 3 months treatment of HD enough for the evaluation of the survival of the patients?

3. In page 7 lines 1-2, the authors should describe the percentage of missing data and how did the authors perform the statistical analyses for missing data.

4. In page 8 line 18, are there any differences in survival between twice-weekly and triple-weekly HD patients?

5. In page 9, Table 1, the rate of cancer in cause of death is very low. Is there any explanation for it? Were the enrolled HD patients relatively younger?

6. In page 10, Tables 1 and 2, the abbreviations should be indicated in the bottom of the tables.

7. In page 13, Table 3, the HR is calculated by the 1 unit increase or 1 SD increase of variables?

8. In pages 14-16, Tables 4, 5, and 6, the authors can explain why age was not a risk for death in HD patients with diabetes?

9. In page 15 line 7, is the risk of a catheter for mortality shown in the tables? How many patients received the catheter HD?

10. In page 17 line 8, the elevation of ferritin ≥ 800 ng/mL is related to severe inflammation or presence of hemophagocytosis such as in malignant lymphoma and adult-onset Still disease?

11. In page 17 lines 20-23, are the differences in ferritin levels due to differences of dosage of iron supplementation?

Reviewer #2: The authors try to reveal the predictive effects of Hb on mortality. The viewpoints of this study are new and attractive to clinicians. However, the analyses and data are preliminary. The authors need to clarify the following points.

1. It is unclear why the authors select 11g/dl as a cutoff level in this study. Can similar data be shown if the authors select 10 g/dl or 12 g/dl as a cutoff level? This is the very critical point of this study.

2. Around 30% of the cause of death is infection. Inflammation should strongly affect iron metabolism and anemia. This is also a critical point of this study. The authors stated that they have no data about C-reactive protein as a limitation. If so, the result and conclusion are very weak and unacceptable. The authors need to adjust or evaluate the inflammatory factor in this study. A count of white blood cells might be one candidate for that.

3. Control levels of diabetes are a key point for infection and/or cardiovascular events. Therefore, control levels of diabetes, such as glycoalbumin, are required. Moreover, SGLT2 inhibitors participate a good role in cardiovascular events. The data on the usage of SGLT2 inhibitors also require.

4. Microvascular or macrovascular damage is also a key factor in cardiovascular events. These data, such as retinopathy or ABI/PWV(ankle-brachial index/pulse wave velocity), are required.

---

## [Author Response · Author response to Decision Letter 0]

8 Dec 2022

The response to reviewers was attached with the revised manuscript.

---

## [Editor Report · Decision Letter 1]

11 Jan 2023

Anemia biomarkers and mortality in hemodialysis patients with or without diabetes: a 10-year follow-up study

PONE-D-22-23532R1

Dear Dr. Aoun, 

We’re pleased to inform you that your manuscript has been judged scientifically suitable for publication and will be formally accepted for publication once it meets all outstanding technical requirements.

Kind regards,

Donovan Anthony McGrowder, PhD., MA., MSc

Academic Editor

PLOS ONE

Additional Editor Comments:

Dear Dr. Aoun,

 The manuscript entitled “Anemia biomarkers and mortality in hemodialysis patients with or without diabetes: a 10-year follow-up study” was revised in accordance with the reviewers’ comments and is provisionally accepted pending final checks for formatting and technical requirements.

Regards,

Dr. Donovan McGrowder (Academic Editor)

---

## [Editor Report · Acceptance letter]

23 Jan 2023

PONE-D-22-23532R1 

Anemia biomarkers and mortality in hemodialysis patients with or without diabetes: a 10-year follow-up study 

Dear Dr. Aoun:

I'm pleased to inform you that your manuscript has been deemed suitable for publication in PLOS ONE. Congratulations! Your manuscript is now with our production department. 

Kind regards, 

on behalf of

Dr. Donovan Anthony McGrowder 

Academic Editor

PLOS ONE